# Rupture Rate, Functional Outcome and Patient Satisfaction after Primary Flexor Tendon Repair with the Modified 4-Strand Core Suture Technique by Tsuge and Using the Arthrex FiberLoop^®^ with Early Motion Rehabilitation

**DOI:** 10.3390/jcm10194538

**Published:** 2021-09-30

**Authors:** Stephanie Vanessa Koehler, Michael Sauerbier, Athanasios Terzis

**Affiliations:** 1Department of Plastic, Hand and Reconstructive Surgery, BG Trauma Center Frankfurt am Main, 60389 Frankfurt am Main, Germany; athanasios.terzis@bgu-frankfurt.de; 2Private Practice for Hand and Plastic Surgery, 61348 Bad Homburg vor der Höhe, Germany; sauerbier@profsauerbier.com

**Keywords:** Arthrex FiberLoop^®^, primary flexor tendon repair, 4-strand suture technique

## Abstract

Purpose:Our hypothesis was that the rupture rate after primary flexor tendon repair in the modified 4-strand core suture technique using the FiberLoop^®^ (Arthrex, Munich, Germany) is lower than in other suture materials and functional outcome and patient satisfaction are superior compared to the current literature. Patients and methods: A 2-stage retrospective, randomized follow-up study of 143 patients treated with the Arthrex FiberLoop^®^ after flexor tendon injury in zones 2 or 3 from May 2013 to May 2017 was performed. In the 1^st^ stage, the rupture rate of all patients was assessed after a follow-up of at least one year by interview to exclude revision surgery. In the 2nd stage, 20% of the patients could be randomly clinically examined. Functional parameters, such as finger and wrist range of motion measured by goniometer, grip strength measured by Jamar dynamometer (Saehan, South Korea), patient satisfaction measured by school grades (1–6), pain levels measured by visual rating scales (0–10) and functional outcome according to the DASH-score were assessed. The Buck-Gramcko and Strickland scores were calculated. The length of sick leave was recorded. Results: A rupture rate of 2.1% was recorded. 29 patients (20%) were followed up at a mean of 34 ± 7.5 months postoperatively. 10.3% of these patients had an incomplete fingertip palm distance. The mean postoperative grip strength was 24 ± 3.1 kg. 93% of the patients were very satisfied with the treatment. No patient complained of pain postoperatively. The mean postoperative DASH score was 6.7 ± 2.8 points. The mean Buck-Gramcko score was 14 ± 0.2 points. 93% of the patients had excellent and 7% good results according to the Strickland score. 67% of patients had a work accident and returned to work at a mean of 4 ± 0.2 months postoperatively. 31% of patients suffered a non-occupational injury and returned to work at a mean of 3 ± 0.4 months postoperatively. Conclusions: Primary flexor tendon repair in the modified 4-strand core suture technique using the Arthrex FiberLoop^®^ has proven to be a viable treatment option in our series. The rupture rate was lower than in other suture materials. It leads to acceptable pain relief, grip strength and functional outcome. Level of Evidence: IV; therapeutic.

## 1. Introduction

Flexor tendon repair is a cornerstone in hand surgery. Roughly 300,000 tendon operations of the hand are performed each year in the United States [1]. Even though comparable data for the European Union is currently unavailable, tendon repair remains a dominant part of acute hand trauma care. According to the “injury-type” directory of the German occupational health insurance (“Verletzungsartenverzeichnis”), damage to a flexor tendon of the hand is classified as a severe injury because a lack of treatment could lead to longer periods of recovery and ultimately loss of hand function [2,3,4]. The care of primary flexor tendon injury depends not only on the quality of the surgical procedure, but also heavily on postoperative treatment. This still remains a key challenge for hand surgeons, occupational therapists, physiotherapists and patients [3,4]. The main goal of postoperative treatment is to maintain mobility of the sutured tendon whilst implementing as little force as possible. Active flexion of the finger can help heal the sutured tendon by reducing a build-up of adhesions with neighbouring structures, which in turn improves its gliding efficiency [2,3,4,5]. Still, the suture material alone cannot handle the average strain exerted onto the repaired flexor tendon as it could rupture even with the smallest amount of load. Stability is only acquired once the collagen fibrils have been reorganized after 12 weeks [6].

In the literature, there is a plethora of repair techniques, suture materials and postoperative treatment strategies. The greater the stability of the suture, the greater the load that can be applied onto the tendon for early dynamic postoperative care. The suture must hence guarantee sufficient tensile strength without undermining the frictionless gliding of the tendon [2,7]. Current literature mainly supports a 4-strand locking core suture and additionally epitendinous repair [2,8,9]. The tensile strength of the tendon increases proportionally to the number of locking core sutures. Whilst a 2-strand suture has been shown to withstand load for passive postoperative treatment, a 4- or 6-strand suture has demonstrated sufficient tensile strength for active postoperative mobilization [10,11,12,13,14,15,16,17,18,19,20]. Conversely, a greater number of strand sutures could thicken the tendon and restrict its gliding efficiency. 

The rupture rate after primary flexor tendon repair is an important parameter for treatment quality [21]. Even though there is a general agreement on the type of suture technique and postoperative treatment strategy, the optimal suture material remains unclear [22]. Identifying the ideal suture material could help provide sufficient stability for early mobilisation and thus improve functional parameters, such as total active range of motion, grip strength and daily activities without leading to higher rupture rates. 

The purpose of this study was to perform a 2-stage retrospective, randomized follow-up of patients who underwent flexor tendon repair on one finger in zones 2 or 3 with the 4-strand modified suture technique by Tsuge, using the FiberLoop^®^ thread (Arthrex, Munich, Germany) with consequent early dynamic rehabilitation (as shown in Figure 1 and Figure 2). The primary aim was to investigate the rupture rate and postoperative complications as well as the functional outcome and patient satisfaction compared to the current literature.

## 2. Patients and Methods

In our department, 143 adult patients (18 years or older) with flexor tendon injury of one finger in zones 2 or 3 of the hand were treated with the 4-strand modified core suture technique by Tsuge using the Arthrex FiberLoop^®^ and Tsuge suture technique with subsequent early mobilisation between May 2013 and May 2017. A subsequent epitendinous suture repair with a synthetic monofilament suture material, such as Prolene 5/0, was performed in a continuous suture technique. Our postoperative rehabilitation protocol lasts 12 weeks. Early passive and active assisted mobilization begin the first day post-surgery with physiotherapy as well as occupational therapy and a dynamic splint is placed on the patient’s forearm and injured finger. The dynamic splint is worn for 6 weeks postoperatively. Inclusion and exclusion criteria are shown in Table 1. Our postoperative rehabilitation protocol is shown in Figure 3. 

In the 1st stage, the rupture rate in all 143 treated patients was assessed after a minimum follow-up of one year per analysis of our medical records and by phone interview to exclude revision surgery in other hospitals and postoperative complications. Any complications leading to revision surgery were reported. A follow-up of at least one year was implemented in this study as the function of an injured finger can improve for up to one year postoperatively because of the gradually increasing range of motion [21].

In the 2nd stage, all 143 patients were invited for clinical examination and to fill out a survey by mail and telephone. Of these, 29 patients (20%) responded to the invitation. Functional parameters, such as finger range of motion (ROM) measured by goniometer in degrees, grip strength measured by Jamar dynamometer (Saehan, South Korea) in kilograms, subjective patient satisfaction measured by German school grades from 1 (excellent) to 6 (fail), pain levels measured by visual analogue pain scales from 0 (no pain at all) to 10 (the worst pain ever possible), and functional outcome according to the DASH-score from 0 (no disability) to 100 (most severe disability) were assessed. The Buck-Gramcko scores and Strickland scores were calculated and compared to the current literature. The period of sick leave of each patient was also recorded in the group of patients that suffered an occupational and non-occupational injury. The same investigator performed all measurements. 

The range of motion of the injured finger was measured using a goniometer in degrees. The angles of the metacarpophalangeal as well as proximal and distal interphalangeal joints were measured in maximum active extension and flexion, with the forearm and wrist in neutral position.

Grip strength was measured using a Jamar dynamometer at level 2 (Saehan, South Korea). The affected hand was measured three times. The mean strength of these three measurements was calculated. 

Subjective patient satisfaction was measured using German school grades ranging from 1 (excellent) to 6 (fail). The patients were verbally asked how satisfied they were with the surgical outcome. 

Using the visual analogue pain scale, patients were asked to circle the number between 0 and 10 that best fits their pain intensity. Zero represents “no pain at all” and 10 represents “the worst pain ever possible”. Several studies have shown that visual rating scales exhibit high correlations with other pain-assessment tools and have proven to have good feasibility in its use and good compliance [23,24]. 

The Disabilities of the Arm, Shoulder and Hand (DASH) questionnaire was developed by the American Academy of Orthopedic Surgeons and the Institute for Work and Health [25] and translated into German [26]. The DASH evaluates self-reported disability using 30 questions assessing function, pain and symptoms of the upper extremity during the preceding week. The patients record a value between 1 and 5 for each item and a final score is calculated, ranging from 0 (no disability) to 100 (most severe disability).

The Buck-Gramcko method of assessment is a 15-point score based on total flexion of the interphalangeal joints (6 points), total extension lag (3 points), and total active range of motion (6 points) [27]. A score of 14–15 points means excellent tendon outcome, a score of 11–13 translates to good tendon outcome, a score of 7–10 fair tendon outcome and a score of 0–6 poor tendon outcome. 

The Strickland evaluation system is based on a formula substracting the active flexion of the PIP and DIP joints from the extension deficit of the PIP and DIP joints, dividing the answer by 175° and multiplying it by 100% [28]. An excellent functional outcome ranges from 85–100%, a good outcome from 70–84%, a fair outcome from 50–69% and a poor outcome is less than 50%. 

For this study, the necessary and appropriate consent was obtained for each patient and the study protocol conformed to the ethical guidelines of the 1975 Declaration of Helsinki as reflected in a prior approval by the ethics committee of the local state medical association (FF 120/2018). 

### Statistical Analysis

Data, such as the rupture rate, are described using frequency and percentage of the whole collective. Continuous variables with a normal distribution, such as finger mobility, grip strength, length of sick leave, pain levels, subjective patient satisfaction, DASH, Buck-Gramcko and Strickland data are exhibited using mean and standard deviation. A confidence interval of *p* < 0.05 was applied and considered statistically significant. 

## 3. Results

In total, 143 patients with flexor tendon injury in zones 2 or 3 of the hand, which were treated with the Arthrex FiberLoop^®^, Tsuge suture technique and early postoperative protocol between May 2013 and May 2017, were interviewed by telephone. We had a total of 28 injured thumbs and 115 injured fingers. There was a total of 13 out of 143 cases in which the FDP tendon was completely lacerated and the FDS tendon was only injured on either the radial or ulnar aspect. In these cases, the FDP tendon was sutured in a 4-strand modified core suture technique and the lacerated radial or ulnar aspect of the FDS tendon was sutured in a 2-strand core technique. All cases with an FDS tendon injury proximal to the division of the FDS tendon into its radial and ulnar aspects were repaired with a 4-strand modified core technique. No FDS tendon was resected in this study. All computer records were furthermore investigated to exclude any revision surgery and postoperative complications. A total rupture rate of 2.1% was recorded. 

29 patients (20%) responded to the mail and phone invitation for clinical examination in our department. 80% of patients declined participation in the 2nd phase of the study owing to unreachability by telephone, disinterest or living too far away. The 29 patients were clinically examined at a mean of 34 ± 7.5 months postoperatively. 10.3% of these patients had an incomplete fingertip palm distance. The mean postoperative grip strength was 24 ± 3.1 kg. 93% of these patients were very satisfied (school grade 1) with the outcome. No patient complained of pain postoperatively. The mean postoperative DASH score was 6.7 ± 2.8 points. The mean Buck-Gramcko score was 14 ± 0.2 points. According to the Strickland score, 93% of these patients had excellent and 7% good results. 67% of patients had a work accident and returned to work at a mean of 4 ± 0.2 months postoperatively. 31% of patients suffered a non-occupational injury and returned to work at a mean of 3 ± 0.4 months postoperatively. These results are exhibited in Table 2. 

## 4. Discussion

The major improvement in the basic science of flexor tendon repair is the understanding of the multiple factors that may affect the strength of the tendon repair. The core suture material, core suture technique and postoperative rehabilitation protocol affect the repair strength. Lacerated flexor tendons are sutured to reapproximate the tendon ends and permit healing. A successful repair of a flexor tendon provides strength, permits glide and results in reduced adhesions. Recent focus has been to increase repair strength in response to more rigorous rehabilitation techniques with early active mobilisation being currently accepted as the ideal rehabilitative method [21]. Unfortunately, two of the major factors over which we have control in our treatment of flexor tendon injury, strength of repair and mobilisation, are at odds with each other regarding the risk of rupture and adhesion formation. Those that contribute to rupture discourage adhesion formation and vice versa. Early mobilisation of the affected finger is paramount to obtain good functional glide, and yet if the strength of the surgical repair is inadequate the risk of rupture is increased. Yet, without early active mobilisation rupture of the tendon is less likely, but adhesions are more likely to form. As a result, focus on flexor tendon repair is on having a high repair strength in order to withstand early active mobilisation and avoid both rupture and adhesion formation [21]. 

The main finding of this study was that after a minimum follow-up of one year the overall rupture rate in our series was 2.1% using the Arthrex FiberLoop^®^, which is lower than that of studies using other suture materials. According to the literature the average rupture rate after flexor tendon repair using other suture materials lies at 6–17% [5,21]. A reduced postoperative rupture rate leads to less revision surgeries and consequently a better prognosis of hand function. 

Furthermore, functional outcome of the affected finger in our series is comparable to the current literature according to the Buck-Gramcko and Strickland scores [16]. Flexor tendon repair with the Arthrex FiberLoop^®^ and 4-strand modified Tsuge technique has shown to lead to acceptable pain relief as well as grip strength. Furthermore, patient satisfaction rates remain high. In the literature excellent or good function has been reported in 70–80% of fingers after primary flexor tendon repair [16,29].

As most studies report similar functional results with other suture materials at one year follow-up or more, it is important to consider the burden of complications following tendon repair, such as rupture and adhesion formation. There were three patients in this study (2.1%) that had a rupture following flexor tendon suture and who underwent revision surgery. They reported by telephone a poor functional outcome of the affected finger. They were unfortunately unavailable for follow-up. 

There are some limitations to this study. One limitation was that only 20% of the patients could be included in the clinical examination in the 2nd phase of the study. As in any study, the response rate to participation in the whole study is critical and a higher response rate would have been desirable. One of the main reasons for non-participation in the 2nd part of the study was the patient’s disinterest in visiting our clinic for follow-up examination as they no longer live nearby. Nevertheless, we do not believe that the difference in loss to clinical follow-up affects the overall findings of our study. Another limitation to this study was the varied expertise of the surgeon, which has been accepted as an important factor influencing the outcome of the tendon surgery. In our department, the residents carried out some of the of the operations with the assistance of a senior surgeon [29]. Furthermore, treatment of flexor tendon injuries requires teamwork and active participation of the patient in rehabilitation is mandatory. Compliance is difficult to define and measure, and as such could not be included in our analysis. 

In conclusion, primary flexor tendon repair with the Arthrex FiberLoop^®^ and 4-strand modified technique by Tsuge has proven to be a very reliable treatment option. The rupture rate of 2.1% using the Arthrex FiberLoop^®^ in our series is lower than in other suture materials and the functional outcome of the injured finger is comparable to the current literature according to the Buck-Gramcko and Strickland scores. Yet, more and larger studies of long-term results after flexor tendon repair using the Arthrex FiberLoop^®^ and 4-strand Tsuge suture technique are warranted. A resolution to current flexor tendon repair of the hand may help enable improved functional outcome and a common agreement on a best practice repair suture material.

## Figures and Tables

**Figure 1 jcm-10-04538-f001:**
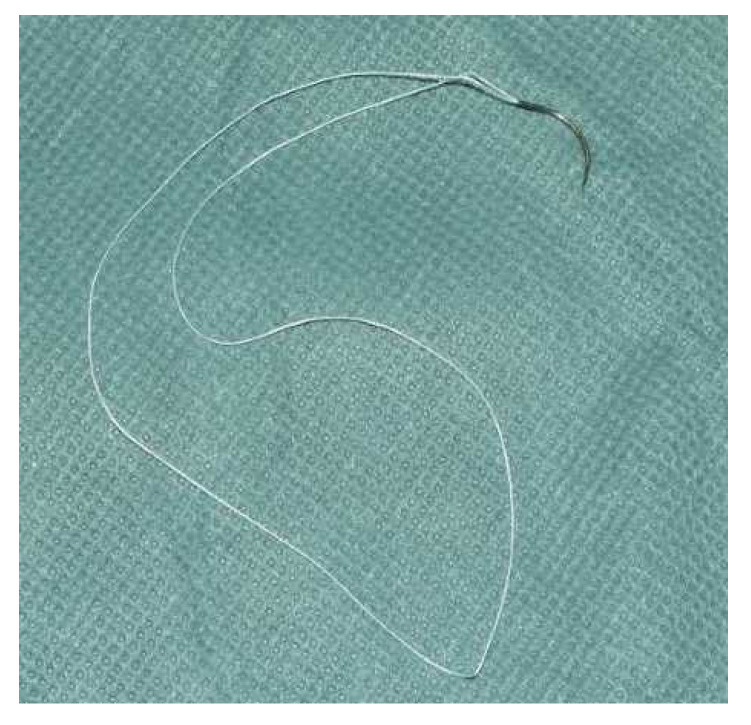
FiberLoop^®^ suture material 4.0.

**Figure 2 jcm-10-04538-f002:**
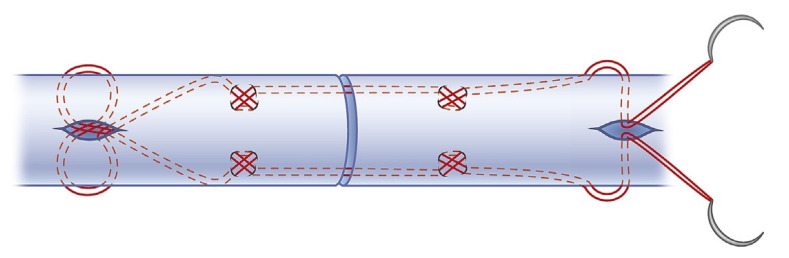
4-strand modified suture technique by Tsuge [14].

**Figure 3 jcm-10-04538-f003:**
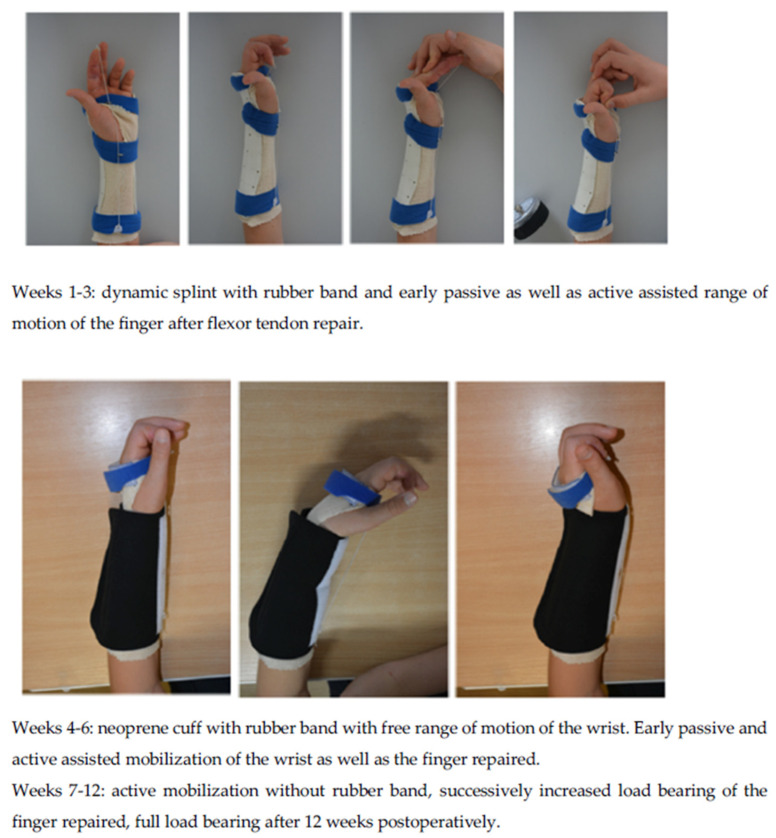
Our 12-week postoperative rehabilitation protocol in 3 steps.

**Table 1 jcm-10-04538-t001:** Inclusion and exclusion criteria for patients after primary flexor tendon repair for selection to a 2-stage retrospective, randomized follow-up study comparing the rupture rate and functional outcome to the current literature.

Inclusion Criteria	Exclusion Criteria
Primary repair of the flexor digitorum profundus and/or flexor digitorum superficialis tendons in zones 2 or 3 of the hand according to the Verdan classification on one finger	Complex hand injury
4/0 Arthrex FiberLoop^®^ for the core suture of the flexor digitorum profundus tendon	Replantation or avascular finger/hand
4-strand core locking Tsuge suture technique	Concomitant fractures of the hand
Early postoperative range of motion	Injury to other fingers of the hand
Follow-up after at least one year postoperatively	Concomitant extensor tendon injury of the hand
>18 years old	Concomitant dislocation of a joint or fracture
Signed informed consent	Previous operations or injuries to the hand

**Table 2 jcm-10-04538-t002:** The results of 29 patients (20%) in the 2nd phase of the study who were followed-up at least one year after flexor tendon repair with the Arthrex FiberLoop^®^, Tsuge suture technique and early postoperative protocol.

Variables	Postoperative
Grip strength (kg)	24 ± 3.1
Pain (VAS 0–10 points)	0
Patient satisfaction (% patients)	93.0
DASH score (0–100 points)	6.7 ± 2.8
Buck-Gramcko score (0–15 points)	14 ± 0.2
Strickland score (% patients with excellent results)	93.0
Period of disability (months) in patients with occupational accidents	4 ± 0.2
Period of disability (months) in patients with non-occupational accidents	3 ± 0.4

## Data Availability

The data presented in this study are available on request from the corresponding author. The data are not publicly available due to patient privacy and the General Data Protection Regulation.

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
