# Peer review of "Rupture Rate, Functional Outcome and Patient Satisfaction after Primary Flexor Tendon Repair with the Modified 4-Strand Core Suture Technique by Tsuge and Using the Arthrex FiberLoop® with Early Motion Rehabilitation"

_jcm, 2021, doi:10.3390/jcm10194538_

Round 1
Reviewer 1 Report
The main object of the article is to report the rupture rate, functional outcome and patient satisfaction for flexor tendon primary repair, in zones 2 and 3 with Arthrex FiberLoop
The materials of suture for flexor tendon repair is and interesting topic. The population included is quite homogeneous, fractures and multi digits injury are excluded, all tendon injury where treated with the same technique, but some points need to be clarified, questions about surgery technique and rehabilitation need to be answered, see following comments.
The first stage of study takes into consideration a high amount of cases, in this stage the rupture rate is calculated. In stage 2, authors evaluate functional outcome and satisfaction, unfortunately most of patients were lost at follow-up.
At first or second stage, could be much interesting to have information about any other complication, as adhesions, bowstringing, infection, complex regional pain syndrome (CRPS), tenosynovitis and wound dehiscence, patients reported any of these? Do You performed any secondary surgery?
Line – 40 “Flexor tendon repair is an elementary topic in hand surgery”, elementary can be misunderstanding, please consider changing with “Flexor tendon repair is a cornerstone of hand surgery”
Line – 87 ”flexor tendon laceration” please consider change laceration with injury
Line -87 to 91:
Considering 143 cases how many are thumbs and how many fingers?
Flexors digitorum superficialis were repaired in any case? If were repaired, with how many strands? Or was flexors digitorum superficialis cut in any case?
About flexors digitorum profundus repair, do You perform epitendinous (peripheral) suture? Please describe
Do you usually perform venting of pulleys? Please describe
After how many days rehabilitation start? This is performed with passive, active assisted or active mobilization? Where the indication the same for all patients? In case of nerve injury and repair how do you manage the mobilization? The patients wear cast after surgery? Patients use to wear also splints constructed by physiotherapist? For how long? This group of questions are important related to rupture rate.
Line 113 and line 134: “The angles of the metacarpophalangeal as well as proximal and distal interphalangeal joints were measured” and “The Buck-Gramcko method of assessment is a 15-point score based on total flexion of the interphalangeal joints (6 points), total extension lag (3 points), and total active range of motion (6 points).” About Buck Gramcko score, metacarpal phalangeal joint was considered in fingers ROM? Or not?
In table 1 comments:
Flexor lesion associated with digital nerve injury where included? Eventually please consider to count how many cases associated with nerve injury where included in first or second stage.
3/0 Fiber loop was ever used?
Photographs: I suggest one or two photographs about surgery or post-operative
Author Response
I would like to thank you for giving me the opportunity to submit a revised draft of the manuscript titled "Rupture rate, functional outcome and patient satisfaction after primary flexor tendon repair with the modified 4-strand core suture technique by Tsuge and using the Arthrex FiberLoop® with early motion rehabilitation". I appreciate the time and effort that you have dedicated to providing your valuable feedback on this paper. I am grateful for your insightful comments. I have been able to address all the concerns you have raised and to incorporate changes to reflect the majority of the suggestions provided by you. I have highlighted the changes as shown in the attached documents.

Reviewer 2 Report
definitely is interesting promoting the idea about 4 core suture is strong enough to start an early rehabilitation protocol. I suggest explaining in detail about the rehabilitation protocol used, and also the fact the response rate during the second stage was very low that is a weakness. there are other studies out there enforcing the same principle and there are other types of sutures with the same design. I suggest that since this is not an Arthrex funded study the main message should be promoting a looped suture and not the Arthrex suture
Author Response
I would like to thank you for giving me the opportunity to submit a revised draft of the manuscript titled "Rupture rate, functional outcome and patient satisfaction after primary flexor tendon repair with the modified 4-strand core suture technique by Tsuge and using the Arthrex FiberLoop® with early motion rehabilitation". I appreciate the time and effort that you have dedicated to providing your valuable feedback on this paper. I am grateful for your insightful comments. I have been able to address all the concerns you have raised and to incorporate changes to reflect the majority of the suggestions provided by you. I have highlighted the changes as shown in the attached document.

Round 2
Reviewer 1 Report
About “Author: Thank you for pointing this out. No nerve injury, artery injury, fracture was included I this study. Only flexor tendon injuries were included.”
If cases with “digital nerve injury” were excluded from this study, please, write in table under exclusion criteria